# Development of a Metastatic Uveal Melanoma Prognostic Score (MUMPS) for Use in Patients Receiving Immune Checkpoint Inhibitors

**DOI:** 10.3390/cancers13143640

**Published:** 2021-07-20

**Authors:** Deirdre Kelly, April A. N. Rose, Thiago Pimentel Muniz, David Hogg, Marcus O. Butler, Samuel D. Saibil, Ian King, Zaid Saeed Kamil, Danny Ghazarian, Kendra Ross, Marco Iafolla, Daniel V. Araujo, John Waldron, Normand Laperriere, Hatem Krema, Anna Spreafico

**Affiliations:** 1Princess Margaret Cancer Center, Division of Medical Oncology and Hematology, University Health Network, Toronto, ON M5G1Z5, Canada; deirdre.kelly@uhn.ca (D.K.); thiago.muniz@uhn.ca (T.P.M.); david.hogg@uhn.ca (D.H.); marcus.butler@uhn.ca (M.O.B.); sam.saibil@uhn.ca (S.D.S.); marco.iafolla@uhn.ca (M.I.); daniel.araujo@edu.famerp.br (D.V.A.); 2Department of Oncology, McGill University, Montreal, QC H3A1G5, Canada; 3Segal Cancer Centre & Lady Davis Institute for Medical Research, Jewish General Hospital, Montreal, QC H3T1E2, Canada; 4Princess Margaret Cancer Center, Tumor Immunotherapy Program, Toronto, ON M5G1Z5, Canada; kendra.ross@uhn.ca; 5Department of Laboratory Medicine and Pathobiology, University of Toronto, Toronto, ON M5G1Z5, Canada; ian.king@uhn.ca (I.K.); zaid.saeed_kamil@uhn.ca (Z.S.K.); dr.danny.ghazarian@uhn.ca (D.G.); 6Princess Margaret Cancer Center, University Health Network, Toronto, ON M5G1Z5, Canada; 7Department of Medical Oncology, Hospital de Base, Sao Jose do Rio Preto 15090-000, SP, Brazil; 8Princess Margaret Cancer Centre, Department of Radiation Oncology, University of Toronto, Toronto, ON M5G1Z5, Canada; john.waldron@rmp.uhn.ca (J.W.); norm.laperriere@rmp.uhn.ca (N.L.); 9Princess Margaret Cancer Centre, Ocular Oncology Service Krembil Research Institute, University Health Network, Toronto, ON M5G2C1, Canada; hatem.krema@uhn.ca

**Keywords:** uveal melanoma, immunotherapy, immune checkpoint inhibitor, PD1, CTLA4, prognostic, predictive score

## Abstract

**Simple Summary:**

This is a retrospective cohort study of metastatic uveal melanoma patients. This study undertook to identify clinical characteristics that were predictive and prognostics of benefit to immune checkpoint inhibitors in patients with metastatic uveal melanoma. We developed a Metastatic Uveal Melanoma Prognostic risk Score based on retrospective data that is comprised of 3 readily available clinical variables (time to metastatic diagnosis, presence of bone metastases, and LDH). Our findings demonstrated that the Metastatic Uveal Melanoma Prognostic risk Score was associated with a statistically significant association with overall survival outcomes in patients with metastatic uveal melanoma treated with ICI. There was a significant predictive association with disease control to ICI for patients with a ‘good risk’ Metastatic Uveal Melanoma Prognostic risk score. This is one of the larger analysis of clinical outcomes in metastatic uveal melanoma patients to date and could inform clinical decision-making.

**Abstract:**

Metastatic uveal melanoma (mUM) is a rare disease. There are limited data on prognostic clinical factors for overall survival (OS) in patients with mUM treated with immune checkpoint inhibitors (ICI). Retrospective and non-randomized prospective studies have reported response rates of 0–17% for anti-PD1/L1 ± anti-CTLA4 ICI in mUM, indicating a potential benefit only in a subset of patients. This study evaluates the characteristics associated with ICI benefit in patients with mUM. We performed a single-center retrospective cohort study of patients with mUM who received anti-PD1/L1 ± anti-CTLA4 ICI between 2014–2019. Clinical and genomic characteristics were collected from a chart review. Treatment response and clinical progression were determined by physician assessment. Multivariable Cox regression models and Kaplan–Meier log-rank tests were used to assess differences in clinical progression-free survival (cPFS) and OS between groups and identify clinical variables associated with ICI outcomes. We identified 71 mUM patients who received 75 lines of ICI therapy. Of these, 54 received anti-PD1/L1 alone, and 21 received anti-PD1/L1 + anti-CTLA4. Patient characteristics were: 53% female, 48% were 65 or older, 72% received one or fewer lines of prior therapy. Within our cohort, 53% of patients had developed metastatic disease <2 years after their initial diagnosis. Bone metastases were present in 12% of patients. The median cPFS was 2.7 months, and the median OS was 10.0 months. In multivariable analyses for both cPFS and OS, the following variables were associated with a good prognosis: ≥2 years from the initial diagnosis to metastatic disease (*n* = 25), LDH < 1.5 × ULN (*n* = 45), and absence of bone metastases (*n* = 66). We developed a Metastatic Uveal Melanoma Prognostic Score (MUMPS). Patients were divided into 3 MUMPS groups based on the number of the above-mentioned prognostic variables: Poor prognosis (0–1), Intermediate prognosis (2) and Good prognosis (3). Good prognosis patients experienced longer cPFS (6.0 months) and OS (34.5 months) than patients with intermediate (2.3 months cPFS, 9.4 months OS) and poor prognosis disease (1.8 months cPFS, 3.9 months OS); *p* < 0.0001. We developed MUMPS—a prognostic score based on retrospective data that is comprised of 3 readily available clinical variables (time to metastatic diagnosis, presence of bone metastases, and LDH). This MUMPS score has a potential prognostic value. Further validation in independent datasets is warranted to determine the role of this MUMPS score in selecting ICI treatment management for mUM.

## 1. Introduction

Uveal melanoma is the most common primary intraocular malignant tumor in adults. Up to 50% of patients diagnosed with uveal melanoma will develop metastatic disease [1]. Twelve-month OS rates of metastatic uveal melanoma (mUM) of up to 52% were seen in first-line prospective clinical studies [2,3]. Uveal melanoma disseminates hematogenously, and 80–90% of patients with mUM will present with the liver as the first site of metastasis [4,5].

The National Comprehensive Cancer Network (NCCN) guidelines prioritize clinical trial enrollment for patients with mUM [6]. Standard practice for managing mUM, outside of clinical trial enrollment, involves systemic therapy or liver-directed therapies, including surgery when the disease is mainly limited to the liver [7,8,9]. Systemic therapy approaches include the same treatments that have a demonstrated efficacy in cutaneous melanoma, including anti-programmed cell death protein 1 (anti-PD1) and anti-cytotoxic T-lymphocyte antigen-4 (anti-CTLA4) immune checkpoint inhibitors (ICI) [10]. Practice -changing trials for ICI in metastatic cutaneous melanoma have described overall response rates (ORR) ranging from 21 to 61% and median progression-free survival (PFS) ranging from 2.9 to 11.6 months [11]. However, patients with uveal melanoma have been excluded from many of the randomized phase 3 clinical trials that have demonstrated the efficacy of ICI in cutaneous melanoma.

Retrospective studies with anti-PD1/L1 monotherapy in mUM have reported objective response rates (ORR) of less than 5%, median PFS of 3 months (0.75–6.75) with a median OS of 5 months (1–16) [12,13]. In retrospective and prospective studies of patients receiving anti-PD1/L1 + anti-CTLA4, ORR ranges from 11 to 18% with a median PFS of 3–5.5 months and OS of 15–20 months [14,15,16,17]. In mUM patients who received expanded access to anti-PD1/L1 + anti-CTLA4, median OS was not reached at a median follow-up of 17.8 months [18]. Preliminary data on tebentafusp, a bispecific soluble TCR therapeutic which redirects T cells to gp100 positive melanocytic cells, has reported an improved disease control rate (DCR) compared to single-agent anti-PD1 or anti-CTLA4 (median OS 21.7 months versus 12 months HR0.51 (95% CI: 0.37, 0.71) with a median follow up of 14.1 months [19]. However, not all patients are eligible to receive tebentefusp, due to a requirement for HLA-matching. As such, anti-PD1/L1 ± anti-CTLA4 remains an important therapeutic option for pts with mUM [16,20,21]. Anti-PD1/L1 and anti-CTLA4 ICI are associated with severe toxicities in 40–64% of patients [2,15,20], and grade 3–4 toxicities frequently lead to early treatment discontinuation and/or long-lasting toxicities that can significantly impact patients’ quality of life.

In the absence of randomized phase III evidence for OS benefit from anti-PD1/L1 ICI regimens in patients with mUM, there is a clinical need to define high-risk prognostic features in this patient population and to ultimately identify predictive biomarkers of ICI response that could assist in the optimal treatment selection for uveal melanoma patients. Therefore, this retrospective analysis aims to evaluate the clinical, biochemical, and molecular characteristics with prognostic and predictive values in a cohort of mUM patients treated with anti-PD1/L1 ICI either as monotherapy or in combination with anti-CTLA4 antibody.

### 1.1. Study Design

This is a retrospective single-center cohort study. This study was approved by the Research Ethics Board at the University Health Network—Princess Margaret Cancer Centre (REB# 19-5186) and was conducted in accordance with the principles of Good Clinical Practice, the provisions of the Declaration of Helsinki, and other applicable local regulations.

### 1.2. Study Population

The Princess Margaret University Health Network (PM-UHN) Tumor Immunotherapy Program (TIP) and melanoma referral databases were used to identify patients with uveal melanoma who received anti-PD1/L1-based ICI (either alone or in combination with anti-CTLA4) as palliative treatment for advanced disease. Eligible patients were defined as those diagnosed with mUM receiving therapy with anti-PD1/L1 ICI with or without an anti-CTLA4 ICI. Patients could have received prior therapies in any treatment line. Cases were collected from January 2014 and December 2019 with baseline imaging and follow-up data. Demographics, treatment parameters, clinical genomic data, and clinical outcomes of interest were extracted from the original patient records and merged into a central database before analysis.

### 1.3. Data Collection and Treatment Outcomes

Clinical data retrieved from electronic medical records included: age, sex, sites of metastatic disease at treatment initiation, size of metastases, baseline Eastern Cooperative Oncology Group (ECOG) performance status, anti-PD1/L1-based regimen used, toxicity, and previous systemic therapies. Serum lactate dehydrogenase (LDH) and neutrophil to lymphocyte ratio (NLR) at the time of treatment initiation were collected and analyzed for their prognostic value. Genomic data were available for a subset of patients and were collected retrospectively. This included information from multiplex ligation-dependent probe amplification (MLPA) testing, and Next Generation Sequencing (NGS). Impact Genetics MLPA testing was done at the time of primary tumor treatment and was used to estimate the probability of metastatic death after treatment of uveal melanoma [22]. MLPA comprises a set of probes, each hybridizing to a specific genomic sequence where chromosome 3 loss and chromosome 8q gain predict poor prognosis, and chromosome 6p gain is associated with improved outcomes and decreased risk of metastasizing [22,23,24]. Within our cohort, NGS was done on recurrent, metastatic UM using either the UHN Melanoma NGS Panel Version 1.1 or the Oncomine Comprehensive Assay v3 (ThermoFisher, Fisher Scientific, Ottawa, ON, Canada). The UHN Melanoma NGS Panel examined exonic coding regions of *BAP1, BRAF, CDK4, CDK6, CDKN2A, GNA11, GNAQ, KIT, NRAS, EIF1AX*, and *SF3B1*. This methodology employs SureSelect Target Enrichment hybrid capture followed by paired-end sequencing on the Illumina sequencing platform. Variant calls were generated using the UHN Clinical Laboratory Genetics custom bioinformatics pipeline with alignment to genome build GRCh37/hg19, and were assessed using Cartagenia Bench Lab NGS v5. The Oncomine Assay is a targeted, next-generation sequencing (NGS) assay that enables the detection of relevant SNVs, indels, CNVs and gene fusions in 161 cancer-related genes. All of the above genes on the UHN Melanoma NGS panel, with the exception of EIF1AX, were included in this assay. Sequencing was performed on the Ion S5 XL System, with data analysis using Ion Reporter 5.12 (ThermoFisher). Variant annotations were obtained from OncoKB when variants were present in the OncoKB database [25].

The best radiologic response to treatment was assessed independently by study investigators (D.K. and A.A.N.R.). Patients were typically restaged with CT scans or MRI every 12 weeks as part of routine clinical care. Best overall response was calculated using baseline and follow-up imaging assessments made by the study investigators (D.K., A.A.N.R.). Tumor response (TR) was defined as radiological evidence of tumor shrinkage of any lesion in the absence of any new growth, as reported by the radiologist. Progressive disease (PD) was defined as significant tumor growth, according to the treating physician assessment of radiologic imaging. Stable disease (SD) was defined as any response that did not meet the criteria for either TR or PD. RECIST criteria were not employed. Clinical progression was based on physician documentation of disease progression or death in the electronic medical record, according to either clinical or radiologic parameters of progression. Clinical progression-free survival (cPFS) was calculated from initiation of ICI treatment to documented clinical progression, death, or last follow-up, defined as the date of death or the last clinic visit. Study data-lock occurred on 8 September 2020. Patients who were alive without clinical progression at the last follow-up were censored. Overall survival (OS) was calculated from initiation of anti-PD1/L1 ICI to death or last follow-up. The reason for treatment discontinuation was also collected. Data were extracted by a first investigator (D.K.) and were reviewed by a second one (A.A.N.R). Discordances in treatment response evaluation were resolved by consensus.

### 1.4. Data Analysis and Statistics

Differences in patient characteristics between treatment groups were examined using Fisher’s exact test. Student’s *t*-test was used to assess statistically significant differences in the number of cycles of therapy received between treatment groups. Univariable and multivariable Cox models were used to evaluate differences in OS and cPFS. Cox regression models were used to calculate Hazard Ratios (HR) with 95% confidence intervals (CI) and *p*-values for survival analyses. We included all clinical variables in the initial multivariable model. Non-significant variables were removed by backward stepwise selection. The final multivariable models for cPFS or OS included only those variables that were associated with *p* < 0.10 in multivariable analysis. Kaplan–Meier survival curves were compared with the log-rank test. All statistical analyses were performed using Stata v12.

## 2. Results

### 2.1. Patient Characteristics

A total of 71 patients with mUM who received 75 lines of anti-PD1/L1 ± anti-CTLA4 ICI for metastatic disease at our institution were identified (Appendix A). In total, 54 lines of therapy were anti-PD1/L1 monotherapy, and 21 lines included anti-PD1/L1 + anti-CTLA4 (Table 1). Of the 75 lines of treatment given, thirty-eight (51%) were naïve to systemic treatment and received ICI as first-line systemic therapy. At the time of ICI, the median age was 64 (range 34–89) years, and 48% of the cohort were 65 or older at the time of treatment initiation. Within the study population, 40 (53%) patients were female, and the majority (69%) were ECOG PS 1 at the time of ICI (Table 1). Body mass index (BMI) was 25 or higher in 68% of the cohort. Serum LDH was elevated over 1.5 times the upper limit of normal in 40% of patients at baseline. Bone metastases were present in 12%, whereas 96% had liver metastases. Other baseline characteristics are listed in Table 1. Patients who received anti-PD1/L1 + anti-CTLA4 therapy were more likely to have a BMI of 25 or higher than patients who received anti-PD1/L1 monotherapy (*p* = 0.054). There was a trend towards younger age in the group that received anti-PD1/L1 + anti-CTLA4, but this was not statistically significant (*p* = 0.130). The remaining variables were not significantly different between those who received anti-PD1/L1 monotherapy and anti-PD1/L1 ± anti-CTLA4.

### 2.2. ICI Treatment Duration and Toxicity

In our entire cohort, 72 (96%) treatments were discontinued at the time of last follow-up. There were no treatment-related deaths that occurred during the observation period. The majority, 61 (81%), discontinued treatment due to disease progression, and 9 (12%) discontinued therapy due to toxicity. Of the 75 lines of therapy given, two had their care transferred to an outside hospital and were lost to follow-up, and three are still receiving ICI treatment (Appendix A). Of the 21 patients who received anti-PD1/L1 + anti-CTLA4, 17 (81%) received high dose anti-CTLA4 (3 mg/kg) with low dose (1 mg/kg) anti-PD1/L1 Q3W followed by anti-PD1/L1 monotherapy at standard doses, and 4 (19%) received low dose anti-CTLA4 (1 mg/kg) combined with standard-dose anti-PD1/L1 (2–3 mg/kg) Q3W followed by anti-PD1/L1 monotherapy at standard doses (Appendix A). Only 7 (33%) patients who started anti-PD1/L1 + anti-CTLA4 received all 4 pre-planned doses (Appendix A). Most patients (57%) who started anti-PD1/L1 + anti-CTLA4 treatment did not go on to receive any anti-PD1/L1 monotherapy. The median number of treatment cycles that included anti-PD1 monotherapy given at standard doses was higher (4 cycles, range 1–38) than for patients who received anti-PD1/L1 + anti-CTLA4 (2 cycles, range 0–29, *p* = 0.0079). Patients who received anti-PD1/L1 + anti-CTLA4 were more likely to have discontinued therapy due to toxicity (4/21; 19%) than patients who received anti-PD1/L1 monotherapy (5/54, 9%), but this difference was not statistically significant (Appendix A).

### 2.3. ICI Treatment Outcomes

For the entire cohort of patients, the median cPFS was 2.7 months (Figure 1A), and the median OS was 10.0 months (Figure 1B). Of these patients, 10 (13%) had a treatment response (TR), 15 (20%) had SD and 49 (65%) had PD as best response. One patient was lost in follow-up before they could be evaluated for treatment response. Treatment responses were observed in 11% of patients who received anti-PD1/L1 monotherapy and 19% of patients who received anti-PD1/L1 + anti-CTLA4. The differences in the treatment response rate were not statistically significant (*p* = 0.589) (Figure 2A). ICI regimen type was not associated with significant differences in cPFS (Figure 2B) or OS (Figure 2C).

### 2.4. Genomic Analyses of Study Population

In this study, of the 75 lines of therapy given, 16 (22.5%) had Impact Genetics chromosomal testing on primary tumor samples, and 50 (70%) received NGS testing on metastatic specimens. Molecular characteristics were heterogeneous across this study and differed between the group of patients who received anti-PD1/L1 alone and those who received anti-PD1/L1 + anti-CTLA4. Chromosome 6p disomy was observed in 33% of lines of anti-PD1/L1 + anti-CTLA4 and 9% of antiPD1/L1 monotherapy (*p* = 0.049), although only a minority of tumors was tested in both groups. As expected, all tested mUM tumors were *BRAF* and *NRAS* wild-type. Amongst lines of anti-PD1/L1 monotherapy, 32 (45%) had tumors tested for NGS; *GNA11* and *GNAQ* mutations were present in 24% and 21%, respectively. Patients who were treated with anti-PD1/L1 + anti-CTLA4 were more likely to have had NGS (86%, *p* = 0.032). In this group, *GNA11* and *GNAQ* mutations were present in 19% and 52% of patients, respectively. As such, *GNAQ* mutations were more common in this group (*p* = 0.024). Similarly, *BAP1* mutations were more commonly identified amongst patients who received anti-PD1/L1 + anti-CTLA4 (54%) compared to those who received anti-PD1/L1 monotherapy (15%, *p* = 0.003). *SF3B1* mutations were observed with similar frequencies in both treatment groups (10% vs. 7%, respectively) (Appendix A). *GNA11* and *GNAQ* mutations were mutually exclusive in our population, and each individual gene was examined in relation to clinical outcomes with no single gene significantly associated with cPFS and OS (data not shown).

### 2.5. Identification of Clinical Variables Associated with Survival

We employed univariate and multivariate cox-regression analyses to assess whether any clinical variables were associated with cPFS or OS in our cohort of mUM patients treated with anti-PD1/L1 ± anti-CTLA4 ICI. Clinical variables that were associated with shorter cPFS in multivariate analyses were: time from the initial diagnosis to metastatic disease ≤2 years (*p* = 0.001), presence of bone metastasis (*p* = 0.086), LDH ≥ 1.5 times the upper limit of normal (x ULN) (*p* = 0.002) and neutrophil-lymphocyte ratio (NLR) > 4 (*p* = 0.005) (Table 2). Clinical variables associated with shorter OS in multivariate analysis were: time from initial diagnosis to metastatic disease of less than 2 years (*p* < 0.001), presence of bone metastasis (*p* = 0.016), liver metastasis with a diameter ≥ 6 cm (*p* = 0.006), LDH ≥ 1.5 × ULN (*p* < 0.001) (Table 3).

### 2.6. Prognostic Value of MUMPS Score

We sought to develop a prognostic score that would have clinical utility for prospectively identifying a cohort of mUM patients that were most likely to have good survival outcomes. MUMPS included only those “good risk” clinical variables that were associated (*p* < 0.10) with both improved cPFS (Table 2) and improved OS (Table 3) in multivariable analyses. These variables were: (i) time from the initial diagnosis to metastatic disease ≥ 2 years; (ii) absence of bone metastases; (iii) LDH < 1.5 × ULN. Patients were divided into risk groups according to the number of MUMPS factors that they possessed: Good risk (3); Intermediate risk (2); and poor risk (0–1). We found that 29%, 51%, and 20% of patients belonged to the good, intermediate, and poor risk groups, respectively (Figure 3A, Appendix A). Patients classified as having good risk also had significantly longer median cPFS (6.0 months) compared to those with intermediate (2.3 months) or poor risk (1.8 months) (*p* = 0.0001, Figure 3C). Good risk patients experienced significantly longer median OS (34.5 months) compared to patients with intermediate (9.4 months) or poor risk scores (3.9 months) (*p* < 0.0001, Figure 3D).

### 2.7. Predictive Value of MUMPS Score

Next, we assessed whether the MUMPS groups could predict benefits from either anti-PD1/L1 monotherapy or anti-PD1/L1 + anti-CTLA4. Patients classified as having a good risk based on their MUMPS prognostic group category were more likely to have achieved TR or SD as the best clinical treatment response and were least likely to have achieved PD as the best clinical treatment response compared to patients with intermediate or poor-risk prognosis, *p* = 0.010 (Figure 3B). For patients with a MUMPS good risk score, there was a non-significant trend (*p* = 0.0750) toward longer cPFS among those who received anti-PD1/L1 monotherapy compared to anti-PD1/L1 + anti-CTLA4 (Figure 4A). MUMPS good-risk patients treated with anti-PD1/L1 monotherapy experienced significantly longer OS (45.3 months) compared to patients with a good prognosis who received anti-PD1/L1 + anti-CTLA4 (15.6 months) *p* = 0.0042 (Figure 4B). There was no significant difference in cPFS (*p* = 0.456) (Figure 4C) or OS (*p* = 0.756) (Figure 4D) according to anti-PD1/L1 treatment regimen among patients with intermediate or poor-risk prognosis.

## 3. Discussion

In contrast to cutaneous melanoma, ICIs have limited activity in mUM, and the standard of care ICI regimen in this setting has not formally been established. Indeed, we observed no significant differences in treatment outcomes when comparing anti-PD1/L1 monotherapy to anti-PD1/L1 + anti-CTLA4 in our entire cohort. Enrollment in clinical trials is an NCCN guideline recommendation for patients with mUM, and prognostic models are needed to better understand the natural history of this rare disease, stratify patients enrolling in clinical trials, and facilitate treatment discussions. In this study, we describe a retrospective cohort of 71 mUM patients who received 75 lines of anti-PD1/L1 ± anti-CTLA4 ICI, and we undertake a comprehensive analysis of clinical variables associated with outcome. The median cPFS (2.7 months) and OS (10.0 months) that we observed in this study were similar to established benchmarks for mUM patients receiving systemic therapy (3.3 and 10.2 months, respectively) [3]. This suggests that our cohort is reasonably representative of the greater mUM patient population. In addition, based on multivariable analyses, we developed a prognostic score (MUMPS) associated with clinically meaningful differences in survival outcomes among a cohort of mUM patients.

The Metastatic Uveal Melanoma Prognostic Score (MUMPS) developed in this study comprises one laboratory value and two clinical variables associated with good outcomes. These variables are readily accessible for assessment by treating physicians and include: ≥ 2 years from the initial diagnosis to metastatic disease, presence of bone metastases and LDH < 1.5 × ULN. Each of these clinical variables has previously been identified in multiple independent studies of survival outcomes in metastatic uveal melanoma patients, and each have a demonstrated prognostic value, irrespective of treatment type [3,14,26]. The clinical prognostic factors in our MUMPS model may be reflective of patients with less aggressive tumor biology. Indeed, short time to relapse and elevated LDH are factors that have been previously reported in other disease sites and may be due to increased tumor burden, aggressive tumor biology, and paranoplastic processes [27,28,29,30].

Bone metastases are also associated with poor prognosis in many other metastatic cancers, including non-small cell lung cancer, urothelial cancer, head and neck cancer, and renal cell carcinoma (RCC) [31,32,33,34]. The presence of bone metastasis may represent a high volume, treatment-resistant biology. It may also reflect the more advanced cases at the time of ICI initiation, where metastasis had extended beyond the liver in a cohort of patients where the majority already have liver involvement. The reasons for its significant association with poorer OS are not fully understood. They may reflect an increase in paraneoplastic processes such as hypercalcemia or poor quality of life and/or functional status among patients with painful bone metastases that limit further systemic therapy interventions or clinical trial enrolment.

Risk stratification and employment of prognostic criteria have proven beneficial for treatment selection in other disease sites. For example, the International Metastatic Renal Cell Cancer Database Consortium (IMDC) criteria [29] is a clinically useful prognostic risk score comprised of clinical variables associated with OS in metastatic RCC. Interestingly, although this IMDC prognostic score was derived from a group of patients who all received anti-angiogenic targeted therapies, it also has clinical utility in predicting the therapeutic benefit of immune checkpoint inhibitors [35]. Similarly, we found that our MUMPS prognostic score may also have a predictive value. We showed that, in our cohort, patients with MUMPS good prognosis derived more benefit from anti-PD1/PDL1 alone vs. anti-PD1/L1 + anti-CTLA4. However, amongst patients with intermediate and poor prognostic MUMPS scores, there was no clear benefit associated with either anti-PD1/L1 regimen. These observations require independent prospective validation before clinical implementation.

Based on predominantly retrospective data and single-arm prospective studies, anti-PD1/L1 + anti-CTLA4 is generally the preferred regimen due to relatively higher rates of reported objective responses compared to anti-PD1/L1 monotherapy in cross-study comparisons [16,20]. Our data challenges the belief that anti-PD1/L1 + anti-CTLA4 should be the preferred systemic ICI for all mUM patients. Indeed, we observed no significant differences in the clinical response rate, cPFS, or OS amongst patients who received anti-PD1/L1 + anti-CTLA4 versus anti-PD1/L1 monotherapy. In our cohort, patients with good prognostic MUMPS scores survived longer with anti-PD1 monotherapy than with anti-PD1/L1 + anti-CTLA4. This result was surprising, but there could be several biologic rationales that would explain this observation. Patients who received anti-PD1/L1 + anti-CTLA4 were more likely to discontinue treatment due to toxicity and received fewer cycles of treatment that contained standard dose anti-PD1/L1 ICIs. It is also possible that increased receipt of subsequent immunosuppressive therapy due to immune-related adverse effects in the anti-PD1/L1 + anti-CTLA4 group may mitigate the antitumor effects of ICI.

In our study, most patients who received anti-PD1/L1 + anti-CTLA4 either progressed early or developed treatment related AEs and did not ever go on to receive anti-PD1/L1 monotherapy maintenance. Responses to anti-PD1/L1 are not as long-lasting in mUM compared to cutaneous melanoma [13,36], and in mUM even complete responses are not necessarily durable for the long term [2]. We speculate that increased duration of exposure to full dose anti-PD1 may be particularly beneficial in mUM; however, this would require further study in independent datasets. Despite this observation, the reported response rates of 12–18% among patients who received anti-PD1 + anti-CTLA4 in prospective trials makes anti-PD1 + anti-CTLA4 a compelling therapeutic option [2,20].

It would be of interest to assess whether anti-PD1 + anti-CTLA4 with lower dose anti-CTLA4 might represent an alternative option to maximize the therapeutic benefit with less dose-limiting toxicity than standard anti-PD1 + anti-CTLA4 dosing regimens. Indeed, low dose anti-CTLA4 (1 mg/kg) with full dose anti-PD1—either 3 mg/kg or 2 mg/kg has been investigated for the treatment of cutaneous melanoma in Checkmate-511 [37] or KEYNOTE-029 [38]. These combination regimens with low dose anti-CTLA4 appear to have similar efficacy to combination ICI with high dose anti-CTLA4 and have been reported to have less serious adverse events than high dose ICI anti-PD1 + anti-CTLA4 and may result in less treatment discontinuation due to toxicity and less requirement for immunosuppressive therapy. In addition, novel CTLA4-free combinations such as Relatlimab, a human IgG4 LAG-3-blocking antibody in combination with nivolumab may also play a role in uveal melanoma treatment in the future [39].

In cutaneous melanoma, it is well established that responses can be maintained for an extended period of time after ICI treatment discontinuation [40]. However, the fundamental underlying biology of uveal melanoma is different from cutaneous melanoma [41], and we do not yet know if ICI responses are maintained after treatment discontinuation in this population. Indeed, continuous treatment with anti-PD1 ICI beyond a year is associated with longer OS compared to a 1-year fixed duration of treatment in non-small cell lung cancer [42]. In a separate, prospective study of 35 mUM patients who received anti-PD1 + anti-CTLA4, discontinuing treatment due to toxicity was not associated with longer PFS [20].

More research is needed to better understand the consequences of holding or discontinuing ICI treatments due to immune-mediated severe adverse events in uveal melanoma, as this may influence further anti-PD1 + anti-CTLA4 drug development, as well as clinicians’ readiness to re-institute ICI after an adverse event. It could be speculated that for some patients with good prognostic features (i.e., those with MUMPS good prognosis disease), the risk of discontinuing or holding treatment early and increased use of immunosuppressive therapy due to increased toxicity with anti-PD1 + anti-CTLA4 vs. anti-PD1 monotherapy may out-weigh the benefits of potentially being more likely to obtain an objective response. Additional studies are required to validate this hypothesis.

Current experimental strategies focus on various single-agent and combination strategies targeting: Melanocyte protein PMEL/GP100, Procaspase-Activating Compound 1, NTRK Inhibitors, LAG-3 Antibodies, Focal Adhesion Kinase (FAK) Inhibitors, Protein kinase C Inhibitors and MEK Inhibitors (Appendix A) [43]. Radioembolization and immunoembolization continue to be studied in combination with anti-PD1 +/- anti-CTLA4. Alternate immune-modulating approaches using autologous tumor-infiltrating lymphocytes (TILS) and T cell receptor (TCR) activation are being explored. Autologous TILS have shown evidence of tumor regression in mUM as well as responses in patients refractory to ICI and IMCgp100, a bispecific molecule that targets the T-cell receptor, has shown antitumor activity and remains in development [44,45].

This study is subject to a number of limitations. This retrospective study analyzed outcomes for patients treated in a single academic center in Canada; therefore, the results may not be generalizable to the broader community. Although this is one of the largest cohort studies to be published for ICI-treated mUM patients, the sample size was still relatively small and therefore subject to bias. For example, the majority (54%) of patients who received single-agent anti-PD1/L1 were 65 years or older, whereas the majority of patients who received anti-PD1 + anti-CTLA4 (67%) were <65 years old. Such unbalance in the patient population may reflect a treatment selection bias by the treating physicians. The response was defined based on investigator interpretation of radiological evidence showing tumor shrinkage and no new metastatic sites.

Another limitation of this study was that genomic profiling was not available for all patients. We have previously established that the presence of oncogenic driver mutations can predict ICI response in cutaneous melanoma [46]. However, due to the smaller sample size and missing genomic data—we could not assess the relationship between genomic driver mutations and ICI response in this cohort of mUM patients. Indeed, the difference in the incidence of *BAP1* mutations between the anti-PD1/L1 and the anti-PD1 + anti-CTLA4 group may be particularly relevant. *BAP1* is a multifunctional tumor suppressor, and *BAP1* mutations in UM are associated with activation of regulatory immune cells and an imunosuppressive tumor microenvironment [47,48]. This could contribute to resistance to ICI and differences in clinical outcomes; however, this is not well understood and is still being elucidated. Genomic analysis was not performed in the majority of patients included in this analysis; therefore, we did not assess whether the inclusion of genomic variables would provide additional prognostic value to our model.

## 4. Conclusions

In summary, we present a prognostic model (MUMPS), comprised of three readily available clinical parameters, which could stratify mUM patients according to their expected prognosis and may have value in predicting benefit from specific ICI regimens. This study provides additional information for health care providers regarding the risk-benefit profile of anti-PD1 + anti-CTLA4. External validation of these data is warranted prior to clinical implementation.

## Figures and Tables

**Figure 1 cancers-13-03640-f001:**
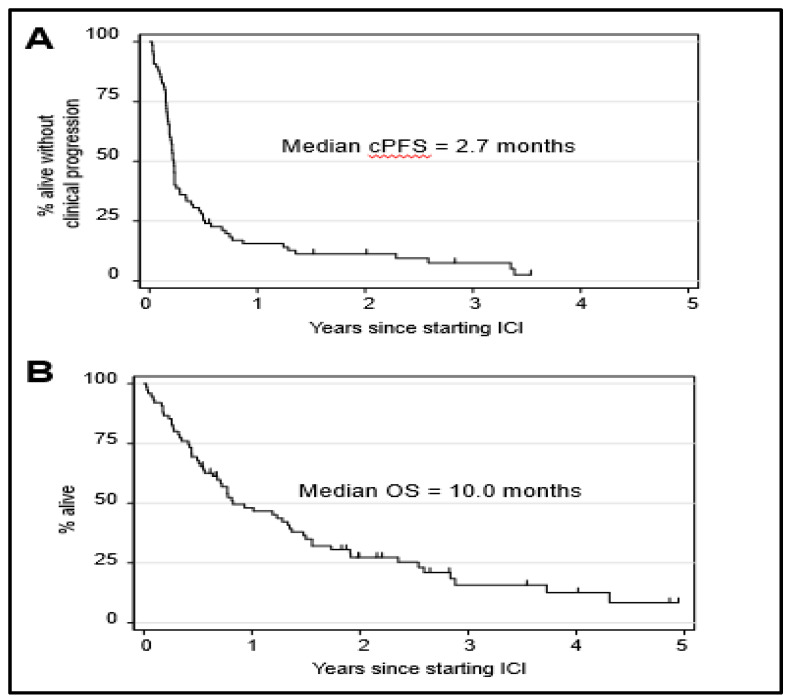
Survival of metastatic uveal melanoma (mUM) patients who received anti-PD1± anti-CTLA4 ICI. (**A**) Clinical progression-free survival and (**B**) overall survival of the entire cohort of 75 mUM patients. Tick marks indicate censored patients.

**Figure 2 cancers-13-03640-f002:**
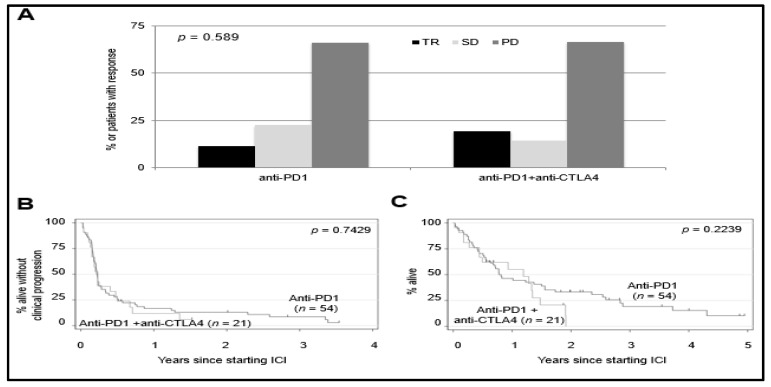
Treatment response and survival outcomes according to anti-PD1 regimen. (**A**)Treatment response according to whether patients received anti-PD1 monotherapy (*n* = 54) or anti-PD1 + anti-CTLA4 (*n* = 21). TR = tumor response, SD = stable disease, PD = progressive disease. The difference in the distribution of treatment responses according to ICI regimen type was assessed with a Fisher’s exact test (*p* = 0.589). (**B**) Clinical progression-free survival and (**C**) overall survival according to anti-PD1 ICI regimen. Log-rank test *p*= 0.7429 (cPFS); *p* = 0.2239 (OS). Tick marks indicate censored patients.

**Figure 3 cancers-13-03640-f003:**
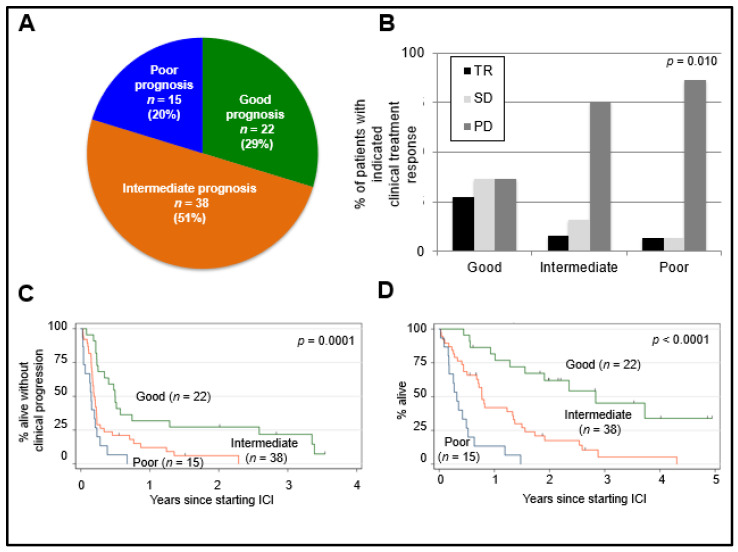
Association between Metastatic Uveal Melanoma Prognostic Score (MUMPS) groups and clinical outcomes. Patients were categorized based on the number of MUMPS good prognosis factors they possessed. Good prognosis factors were absence of bone metastases, >2 years from initial diagnosis to metastatic disease, LDH < 1.5 × ULN. (**A**) The percentage of patients classified as having good prognosis (MUMPS = 3), intermediate prognosis (MUMPS = 2), and poor prognosis (MUMPS = 0–1) are indicated. (**B**) Distribution of clinical treatment responses according to MUMPS groups are indicated; *p* = 0.010; Fisher’s exact test. (**C**) cPFS according to MUMPS groups. (**D**) OS according to MUMPS groups. Log-rank test *p* = 0.0001 (cPFS); *p* < 0.0001 (OS). Tick marks indicate censored patients. Good prognosis (*n* = 22) = green line, Intermediate prognosis (*n* = 38) = orange line, Poor prognosis (*n* = 15) = blue line. TR = tumor response, SD = stable disease, PD = progressive disease.

**Figure 4 cancers-13-03640-f004:**
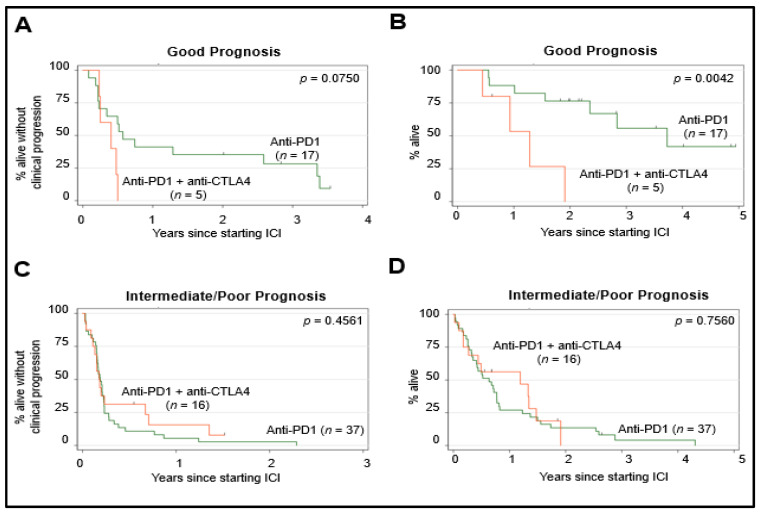
Survival according to MUMPS groups and ICI regimen. Survival outcomes of patients with good prognosis (MUMPS = 3) were stratified according to the anti-PD1/L1 regimen they received: anti-PD1/L1 monotherapy (*n* = 17), anti-PD1 + anti-CTLA4 (*n* = 5). cPFS is Scheme 0. (cPFS); *p* = 0.0042 (OS). Survival outcomes of patients with intermediate (MUMPS = 2) or poor prognosis (MUMPS = 0–1) were stratified according to the anti-PD1 regimen they received: anti-PD1/L1 monotherapy (*n* = 37), anti-PD1 + anti-CTLA4 (*n* = 16). cPFS is shown in (**A**,**C**) and OS is shown in (**B**,**D**). Log-rank test *p* = 0.4561 (cPFS); *p* = 0.7560 (OS).

**Table 1 cancers-13-03640-t001:** Clinical characteristics of 71 mUM patients who received 75 lines of anti-PD1/L1 +/− anti-CTLA4 ICI.

Clinical Characteristics	Entire Cohort (*n* = 75)	Anti-PD1 (*n* = 54)	Anti-PD1 + Anti-CTLA4 (*n* = 21)	* p * -Value
Age				
<65	39 (52%)	25 (46%)	14 (67%)	0.130
≥65	36 (48%)	29 (54%)	7 (33%)	
Sex				
female	40 (53%)	29 (54%)	11 (52%)	1.000
male	35 (47%)	25 (46%)	10 (48%)	
BMI				
<25	24 (32%)	21 (39%)	3 (14%)	0.054
≥25	51 (68%)	33 (61%)	18 (86%)	
ECOG				
0	20 (27%)	13 (24%)	7 (33%)	0.502
1	52 (69%)	38 (70%)	14 (67%)	
2	3 (4%)	3 (6%)	0 (0%)	
Time from dx to metastasis				
<2 years	40 (53%)	32 (59%)	8 (38%)	0.125
≥2 years	35 (47%)	22 (41%)	13 (62%)	
Liver Directed Therapy				
None	66 (88%)	47 (87%)	19 (90%)	
Radiofrequency Ablation	1 (1%)	1 (2%)	0	
Chemoembolization	2 (3%)	1 (2%)	1 (5%)	
Radiation	2 (3%)	2 (4%)	0	
Radioembolization	4 (5%)	3 (6%)	1 (5%)	
# Previous systemic therapies				
0	38 (51%)	30 (56%)	8 (38%)	0.385
1	16 (21%)	10 (19%)	6 (29%)	
≥2	21 (28%)	14 (26%)	7 (33%)	
Prior Immunotherapy				
No	51 (68%)	32 (59%)	9 (42%)	0.302
Yes	24 (32%)	22 (41%)	12 (58%)	
Prior chemotherapy				
No	63 (84%)	47 (87%)	16 (76%)	0.299
Yes	12 (16%)	7 (13%)	5 (24%)	
Liver Metastases				0.188
Absent	3 (4%)	1 (2%)	2 (10%)	
Present	72 (96%)	53 (98%)	19 (90%)	
Lung metastases				
Absent	49 (65%)	36 (67%)	13 (62%)	0.789
Present	26 (35%)	18 (33%)	8 (38%)	
Bone Metastases				
No	66 (88%)	46 (85%)	20 (95%)	0.430
Yes	9 (12%)	8 (15%)	1 (5%)	
Largest metastasis size				
<6 cm	45 (60%)	32 (59%)	13 (62%)	1.000
≥6 cm	30 (40%)	22 (41%)	8 (38%)	
LDH				
<1.5 × ULN	45 (60%)	33 (61%)	12 (57%)	0.797
≥1.5 × ULN	30 (40%)	21 (39%)	9 (43%)	
NLR				
<4	47 (63%)	33 (61%)	12 (57%)	0.476
≥4	28 (37%)	21 (39%)	9 (43%)	

**Table 2 cancers-13-03640-t002:** Clinical variables associated with clinical progression-free survival (*n* = 75).

Clinical Characteristics	Univariable			Multivariable		
	HR	95% CI	*p*-Value	HR	95% CI	*p*-Value
cPFS						
Age > 65	0.99	0.62–1.60	0.984	-	-	-
Male	0.81	0.51–1.30	0.393	-	-	-
ECOG ≥ 1	1.28	0.76–2.18	0.355	-	-	-
BMI ≥ 25	0.64	0.39–1.05	0.079	-	-	-
Time from initial dx to stage IV <2 years	1.82	1.14–3.05	0.016	2.45	1.44–4.18	0.001
>1 prior systemic therapy	0.76	0.38–1.19	0.327	-	-	-
Prior immunotherapy	1.15	0.72–1.86	0.557	-	-	-
Prior chemotherapy	0.78	0.37–1.41	0.450	-	-	-
ICI regimen: PD1 + CTLA4 vs. PD1	1.09	0.64–1.87	0.745	-	-	-
Lung metastases	1.16	0.71–1.92	0.539	-	-	-
Bone metastases	2.13	1.03–4.42	0.042	1.91	0.91–4.00	0.086
Largest metastasis diameter ≥6 cm	1.48	0.92–2.40	0.105	-	-	-
LDH ≥ 1.5 × ULN	2.34	1.43–3.85	0.001	2.26	1.36–3.77	0.002
NLR ≥ 4	1.69	1.05–2.74	0.032	2.14	1.25–3.65	0.005

**Table 3 cancers-13-03640-t003:** Clinical variables associated with overall survival (*n* = 75).

Clinical Characteristics	Univariable			Multivariable		
	HR	95% CI	*p*-Value	HR	95% CI	*p*-Value
Overall Survivial						
Age ≥ 65	1.51	0.90–2.54	0.121	1.69	0.96–2.97	0.068
Male	0.99	0.60–1.66	0.981	-	-	-
ECOG ≥ 1	2.04	1.08–3.89	0.029	-	-	-
BMI ≥ 25	0.74	0.43–1.29	0.297	-	-	-
Time from initial dx to stage IV <2 years	1.86	1.11–3.12	0.019	2.82	1.60–4.95	<0.001
>1 prior systemic therapy	0.83	0.47–1.49	0.541	-	-	-
Prior immunotherapy	1.28	0.76–2.14	0.679	-	-	-
Prior chemotherapy	0.77	0.38–1.58	0.472	-	-	-
ICI regimen: PD1 + CTLA4 vs. PD1	1.44	0.80–2.62	0.227	-	-	-
Lung metastases	1.48	0.87–2.52	0.152	-	-	-
Bone metastases	1.86	0.90–3.87	0.092	2.57	1.19–5.55	0.016
Largest metastasis diameter ≥6 cm	2.11	1.26–3.56	0.005	2.22	1.25–3.92	0.006
LDH ≥ 1.5 × ULN	4.45	2.57–7.73	<0.001	4.25	2.35–7.68	<0.001
NLR ≥ 4	1.65	0.98–2.79	0.062	-	-	-

## Data Availability

The data presented in this study are available on request from the corresponding author.

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
