# Peer review of "Development of a Metastatic Uveal Melanoma Prognostic Score (MUMPS) for Use in Patients Receiving Immune Checkpoint Inhibitors"

_cancers, 2021, doi:10.3390/cancers13143640_

Round 1
Reviewer 1 Report
Major suggested edits:
- Having a short review (in table format) on phase 3 clinical trials studying the efficacy of ICI in cutaneous melanoma can further emphasize the importance of the current study. I suggest highlighting those studies in which patients with uveal melanoma have been excluded.
- The x-axis for Figure-1 should be similar for both panels for comparison purposes
- For Figure-2, the statistical difference (if any) should be marked on panel A. The x-axis for panels B and C should be similar for comparison purposes. Similar considerations apply to Figure-3 and Figure-4. Moreover, acronyms in the figures (e.g. TR, SD, PD) should be mentioned in full format in each figure's legend. The authors are also advised to provide more content on figure legends; it would make it easier for the reader to grasp the take-home message from each figure if presented with more content in figure legends.
- In Table-3, only p-values < 0.05 should be bold indicating the 'significant difference'. Currently, some p-values >0.05 are marked as showing the statistically significant difference, but that does not appear to be the case here.
Minor suggested edits:
- MUMPS risk score should be replaced by MUMPS throughout the manuscript (risk factor is already included in the acronym)
- The last three sentences should be deleted from the abstract (more suitable to have these points in the introduction as they have references too)
- Some instances of inconsistency in the writing should be fixed (e.g. extra spaces throughout the manuscript, phase III > line 82 vs. phase 3 > line 65, etc)
Author Response
Development of a Metastatic Uveal Melanoma Prognostic Risk Score (MUMPS) for use in patients receiving Immune Checkpoint Inhibitors
Deirdre Kelly1*, April A. N. Rose1,2,3*#, Thiago Pimentel Muniz1, David Hogg1, Marcus O. Butler1,4, Samuel D. Saibil1,4, Ian King5,6, Zaid Saeed Kamil5,6, Danny Ghazarian5,6, Kendra Ross4, Marco Iafolla1,4, Daniel V. Araujo1,7, John Waldron8, Normand Laperriere8, Hatem Krema9, Anna Spreafico1,4#
Reviewer 1: Comment #1: Having a short review (in table format) on phase 3 clinical trials studying the efficacy of ICI in cutaneous melanoma can further emphasize the importance of the current study. I suggest highlighting those studies in which patients with uveal melanoma have been excluded.
Author Response: We sincerely thank the reviewer for their feedback. We have updated the summary discussion on clinical outcomes to ICI in cutaneous melanoma is available in the manuscript (lines 66-71). In addition, we have referenced article #11, which provides a comprehensive review of ICI outcomes in cutaneous melanoma, including a table describing key trial results. However, the scope of this research paper primarily focuses on clinical responses in the uveal melanoma setting, therefore we have opted not to include a table of cutaneous melanoma outcomes in our manuscript.
Reviewer 1: Comment #2: The x-axis for Figure-1 should be similar for both panels for comparison purposes. For Figure-2, the statistical difference (if any) should be marked on panel A. The x-axis for panels B and C should be similar for comparison purposes. Similar considerations apply to Figure-3 and Figure-4. Moreover, acronyms in the figures (e.g., TR, SD, PD) should be mentioned in full format in each figure's legend. The authors are also advised to provide more content on figure legends; it would make it easier for the reader to grasp the take-home message from each figure if presented with more content in figure legends.
Author Response: We have corrected this and the figures as suggested.
Reviewer 1: Comment #3: In Table-3, only p-values < 0.05 should be bold indicating the 'significant difference'. Currently, some p-values >0.05 are marked as showing the statistically significant difference, but that does not appear to be the case here.
Author Response: We have updated table 3 as suggested. Only p-values < 0.05 are in bold, indicating the 'significant difference (Line 250).
Reviewer 1: Comment #4: MUMPS risk score should be replaced by MUMPS throughout the manuscript (risk factor is already included in the acronym).
Author Response :We have replaced MUMPS risk score by MUMPS throughout the manuscript (Lines 45,36,53,54,254,256.259,277,278,279,282,283,318,323,341,342)
Reviewer 1: Comment #5: The last three sentences should be deleted from the abstract (more suitable to have these points in the introduction as they have references too).
Author Response: This was a formatting error. The last three sentences from the abstract were adjusted as suggested.
Reviewer 1: Comment #6: Some instances of inconsistency in the writing should be fixed (e.g., extra spaces throughout the manuscript, phase III > line 82 vs. phase 3 > line 65, etc)
Author Response: This was a formatting error. These typos have been corrected throughout as suggested.
Reviewer 2 Report
This retrospective single-centre cohort study based on comprehensive clinical examinations developed a Prognostic Risk Score (MUMPS) system for use in mUM patients receiving immune checkpoint inhibitors.
At risk classification (poor, intermediate, good) I would suggest changing the name of risk to prognostic factor or another name referring to prognosis for better understanding.
According to survival outcomes, the MUMPS shows that the combined therapy (anti-PD1+anti-CTLA4) is not indicated in all cases as opposed to monotherapy. In Good Risk Group in addition to changes in concentrations of anti-CTLA4, what other conditions could be applied to exempt the patient from the co-therapy because of its side-effects?
Since the BMI index (≥25) affects disease and therapy outcome, the question arises whether a non-drastic, controlled diet would positively affect the effectiveness of the therapies against mUM.
Overall, based on ATP III and IDF guidelines the use of waist circumference is more preferred (standard) than BMI. It should be discussed somehow.
Author Response
Reviewer 2: Comment #1: At risk classification (poor, intermediate, good) I would suggest changing the name of risk to prognostic factor or another name referring to prognosis for better understanding.
Author Response: Regarding the MUMPS score, prognostic score is contained in MUMPS acronym. We have removed the word “risk” from our groupings for clarity. The term prognostic factor or score has been added throughout the manuscript for better understanding (line 88, 255, 315, 323, 338,341, 343, 352, 387).
Reviewer 2: Comment #2: According to survival outcomes, the MUMPS score shows that the combined therapy (anti-PD1+anti-CTLA4) is not indicated in all cases as opposed to monotherapy. In Good Risk Group in addition to changes in concentrations of anti-CTLA4, what other conditions could be applied to exempt the patient from the co-therapy because of its side-effects?
Author Response: In terms of individual therapy selection, MUMPS scoring provides a novel classification system and does require validation. Specifically, for the Good Prognosis Group , further research is required to understand if additional demographic or inherent tumor characteristics could be used to assist with therapy selection.
Reviewer 2: Comment #3: Since the BMI index (≥25) affects disease and therapy outcome, the question arises whether a non-drastic, controlled diet would positively affect the effectiveness of the therapies against mUM. Overall, based on ATP III and IDF guidelines the use of waist circumference is more preferred (standard) than BMI. It should be discussed somehow.
Author Response: In our study, we demonstrated that BMI was not significantly associated with differential responses to ICI. We agree that, based on ATP III and IDF guidelines, waist circumference can be an important clinical feature. Unfortunately in our study, waist circumference information was not available and is not routinely used in our clinical practice. Given that BMI was not a significant finding in our study, we have opted not to include a detailed discussion of this in our manuscript.
Reviewer 3 Report
The manuscript presented by Kelly et al, entitled: “Development of a Metastatic Uveal Melanoma Prognostic Risk Score (MUMPS) for use in patients receiving immune check-point inhibitors”, presents a retrospective analysis on the benefit of these therapies in uveal melanoma patients.
Immune check-point inhibitors have limited benefit in UM. The Authors evaluated the clinical and genomic characteristics of UM patients who received PD1/L1 +/- anti CTLA4 treatments. Indeed, they found limited effects and no significant differences in outcome when comparing anti-PD1/L1 monotherapy to anti-PD1/L1+ anti CTLA4.
Nevertheless, the Authors developed a prognostic risk score associated with survival outcomes among a cohort of UM patients, defined as good, intermediate and poor risk. They showed that patients with good risk derived more benefits from anti-PD1/L1 alone, challenging the previous belief that the combination with anti CTLA4 is more effective.
The study is well written, and it presents a valuable prognostic risk score for immune check point therapies. The major weakness is that the genomic analysis was not performed in the majority of the patients, whose characteristics would be essential to better understand the disease and prognostic factors.
Author Response
Reviewer 3: Comment #1: The major weakness is that the genomic analysis was not performed in the majority of the patients, whose characteristics would be essential to better understand the disease and prognostic factors.
Author Response: To Reviewer 3: We appreciate your feedback and acknowledge that genomic analysis was not performed in the majority of the patients. Future collaborative efforts may be able to contribute further insight into genomic differences in advanced uveal melanomas. We have added the following sentences (lines 409-416) to highlight this limitation in our study:
“Another limitation of this study was that genomic profiling was not available for all patients. We have previously established that the presence of oncogenic driver mutations can predict ICI response in cutaneous melanoma[47]. However, due to the smaller sample size and missing genomic data - we could not assess the relationship between genomic driver mutations and ICI response in this cohort of mUM patients. Indeed, the difference in the incidence of BAP1 mutations between the anti-PD1/L1 and the anti-PD1 + anti-CTLA4 group may be particularly relevant. BAP1 is a multifunctional tumor suppressor, and BAP1 mutations in UM are associated with activation of regulatory immune cells and an im-munosuppressive tumor microenvironment[48, 49]. This could contribute to resistance to ICI and differences in clinical outcomes; however, this is not well understood and is still being elucidated.”
Round 2
Reviewer 1 Report
The authors have addressed the issues raised previously.